# CGRP, adrenomedullin and adrenomedullin 2 display endogenous GPCR agonist bias in primary human cardiovascular cells

Ashley J. Clark[1], Niamh Mullooly[2], Dewi Safitri[1,3], Matthew Harris[1], Tessa de Vries[4], Antoinette MaassenVanDenBrink[4], David R. Poyner[5], Davide Gianni[2], Mark Wigglesworth [6] & Graham Ladds [1✉]

Agonist bias occurs when different ligands produce distinct signalling outputs when acting at the same receptor. However, its physiological relevance is not always clear. Using primary human cells and gene editing techniques, we demonstrate endogenous agonist bias with physiological consequences for the calcitonin receptor-like receptor, CLR. By switching the receptor-activity modifying protein (RAMP) associated with CLR we can "re-route" the physiological pathways activated by endogenous agonists calcitonin gene-related peptide (CGRP), adrenomedullin (AM) and adrenomedullin 2 (AM2). AM2 promotes calcium-mediated nitric oxide signalling whereas CGRP and AM show pro-proliferative effects in cardiovascular cells, thus providing a rationale for the expression of the three peptides. CLR-based agonist bias occurs naturally in human cells and has a fundamental purpose for its existence. We anticipate this will be a starting point for more studies into RAMP function in native environments and their importance in endogenous GPCR signalling.

[1] Department of Pharmacology, University of Cambridge, Cambridge, UK. [2] Functional Genomics, Discovery Sciences, R&D, AstraZeneca, Cambridge, UK. [3] Pharmacology and Clinical Pharmacy Research Group, School of Pharmacy, Bandung Institute of Technology, Bandung, Indonesia. [4] Department of Internal Medicine, Erasmus MC, Erasmus University Medical Centre, Rotterdam, Rotterdam, Netherlands. [5] School of Life and Health Sciences, Aston University, Aston Triangle, Birmingham, UK. [6] Hit Discovery, Discovery Sciences, BioPharmaceuticals R&D, AstraZeneca, Alderley Park, UK. ✉email: grl30@cam.ac.uk

G protein-coupled receptors (GPCRs) form the largest protein family in the human genome. Approximately 30% of marketed drugs target these receptors and therefore understanding their signalling pathways is not simply an academic exercise. For many years it had been incorrectly assumed that agonist-occupied GPCRs signalled through a single pathway to elicit their response. However, there is now overwhelming evidence to suggest that many GPCRs exist in multiple receptor conformations and can elicit numerous functional responses, both G protein- and non-G protein-dependent. Furthermore, different agonists, acting at the same receptor have the potential to activate different signalling pathways to varying extents; a concept referred to as biased agonism or signalling bias[1,2]. This can explain why there is apparent duplication amongst endogenous agonists, particularly for peptides. While the therapeutic promise of biased agonists is obvious:[3] it allows design of ligands that actively engage with one beneficial signalling outcome while reducing the contribution from those that mediate more undesirable effects, it is not without controversy. For example, recent doubt has been cast on validity of developing synthetic biased agonists against the μ-opioid receptor—a GPCR considered the trailblazer for therapeutic potential of biased agonism[4]. Thus, further investigations into the role of agonist bias and its physiological importance, particularly its relevance to endogenous agonists, are required to bridge the gap between heterologous studies and in-vivo investigations.

While there are many well-studied GPCRs that exhibit signalling bias, including the aforementioned μ-opioid receptor, we have focused upon the calcitonin-like receptor (CLR) since it couples to multiple G proteins and β-arrestins. Importantly, when co-expressed with one of three receptor-activity modifying proteins, (RAMPs), it can be activated by distinct endogenous agonists; calcitonin-gene related polypeptide (CGRP), adrenomedullin (AM) and adrenomedullin 2/intermedin (AM2). This makes it a good system to investigate the role of bias for endogenous ligands. CGRP, an abundant neuropeptide, is the most potent microvascular vasodilator known. While it is thought to be cardioprotective, it has also been implicated in diseases such as migraine[5]. AM is released by the vascular endothelium and is a potent vasodilator that can modulate vascular tone, it is involved in angiogenesis, and is elevated in some cancers and heart failure[6–8]. AM2 is also a vasodilator and highly expressed in the heart and vasculature[9,10]. It can cause sympathetic activation, have antidiuretic effects, and is upregulated in cardiac hypertrophy and myocardial infarction[11].

Molecularly, CLR and its close relative, the calcitonin receptor, are classical class B1 GPCRs. CLR is pleiotropically coupled, predominately activating $G_s$ although there are reports of couplings to both $G_{i/o}$ and $G_{q/11}$[2,12] families. These Gα subunits promote activation/inhibition of adenylyl cyclase and phospholipase Cβ to generate intracellular second messengers including cAMP and mobilise intracellular $Ca^{2+}$ ($Ca^{2+}_i$) which then activate their respective intracellular signalling cascades. Beyond the Gα subunits CLR has been reported to couple to β-arrestins[13] inducing internalisation, although it has been suggested that this interaction can also lead to its own signalling events possibly promoting cell proliferation[14]. Despite this high potential for agonist-induced pleiotropy, CLR remains most closely associated with adenylyl cyclase activation and generation of cAMP. It is unknown whether this is representative of CLR's true signalling pattern in the endogenous setting.

Additional complexity is added to the pharmacology of the CLR since it has an absolute requirement for the formation of a heterodimer with a RAMP[15,16]. In overexpression studies, each of the three RAMPs has been shown to differentially influence the affinity and agonist bias of the CGRP family of peptides at the CLR[17,18]. CLR in complex with RAMP1 generates the CGRP receptor since CGRP has been demonstrated to be the most potent of the three agonists at this receptor for generation of cAMP. Likewise, CLR-RAMP2 generates the adrenomedullin 1 (AM1) receptor (AM is the most potent at this receptor) and CLR-RAMP3 produces the AM2 receptor (here AM and AM2 are equipotent). To date, the physiological role and cognate receptor for AM2 remain unknown. While there is an abundance of evidence of GPCR signalling bias in recombinant cell systems, and in this case, CLR-mediated bias[12,19], documented examples using natural agonists and endogenously expressed human receptors are currently lacking. We wished to ascertain whether signalling bias at the CLR occurs in primary cells and whether it plays a role in cellular function. We chose to focus on RAMP1 and RAMP2 as the CGRP and AM1 receptors are the best described.

Using human endothelial cells which endogenously express the AM1 receptor, we demonstrate that biased agonism is present and has a fundamental role in the function of peptide hormones acting on primary human cells. Moreover, through deletion of the endogenous RAMP2 and replacing it with RAMP1, we highlight that not only is the RAMP essential for CLR function and CGRP peptide family signalling in primary cell systems but that RAMPs direct the pattern of agonist bias observed. Furthermore, we document previously unreported actions for the CGRP-based peptide agonists; AM2 emerges as an agonist uniquely biased to elevate calcium-mediated nitric oxide (NO) signalling while both CGRP and AM display distinct pro-proliferative effects in cardiovascular cells. The work we describe here reveals that GPCR agonist bias occurs naturally in human cells and plays fundamentally important physiological roles in providing unique functions to endogenous agonists.

## Results

**Endothelial cells exclusively express functional AM1 receptor.** While there are many reports of biased agonism for GPCRs in recombinant systems (e.g.[19–21]), few examples have been documented in primary human cells. Given the reported roles of CGRP, AM and AM2 in the cardiovascular system we have focussed our studies upon these peptides, and their receptors in primary human umbilical vein endothelial cells (HUVECs), a well-established primary human vascular endothelial cell line. Initially, we determined that HUVECs express the AM1 receptor since we could only detect transcripts for *CALCRL* (gene for CLR) and *RAMP2* (Fig. 1a). This was confirmed functionally, since when the endothelial cells were stimulated with agonists and cAMP accumulation quantified (after 30 min stimulation when the response had plateaued (Supplementary Fig. 1a)), the rank order of potency was AM > AM2 > CGRP (Fig. 1b, and Supplementary Table 1). An important factor in confirming receptor-specific agonist bias is to ensure that competing receptors are not present in the system. The closely related calcitonin receptor not only interacts with RAMPs[22] but also binds CGRP with high affinity[23]. HUVECs do not appear to express the calcitonin receptor since we were unable to detect the presence of its transcript (Fig. 1a) or obtain a functional response upon application of two calcitonin receptor agonists (calcitonin or amylin) (Supplementary Fig. 1b). Furthermore, application of the selective AM1 receptor antagonist AM22–52 at 100 nM abolished agonist-induced cAMP accumulation while 100 nM olcegepant (a CGRP receptor-selective antagonist) had little effect (Supplementary Fig. 1c–h). Thus, based upon these data, we suggested that HUVECs specifically express the AM1 receptor alone and are a useful primary cell line with which to study potential endogenous agonist bias.

**Endogenous agonist bias at the AM1 receptor.** For studies of biased agonism, it is not simply the ability of different ligands to

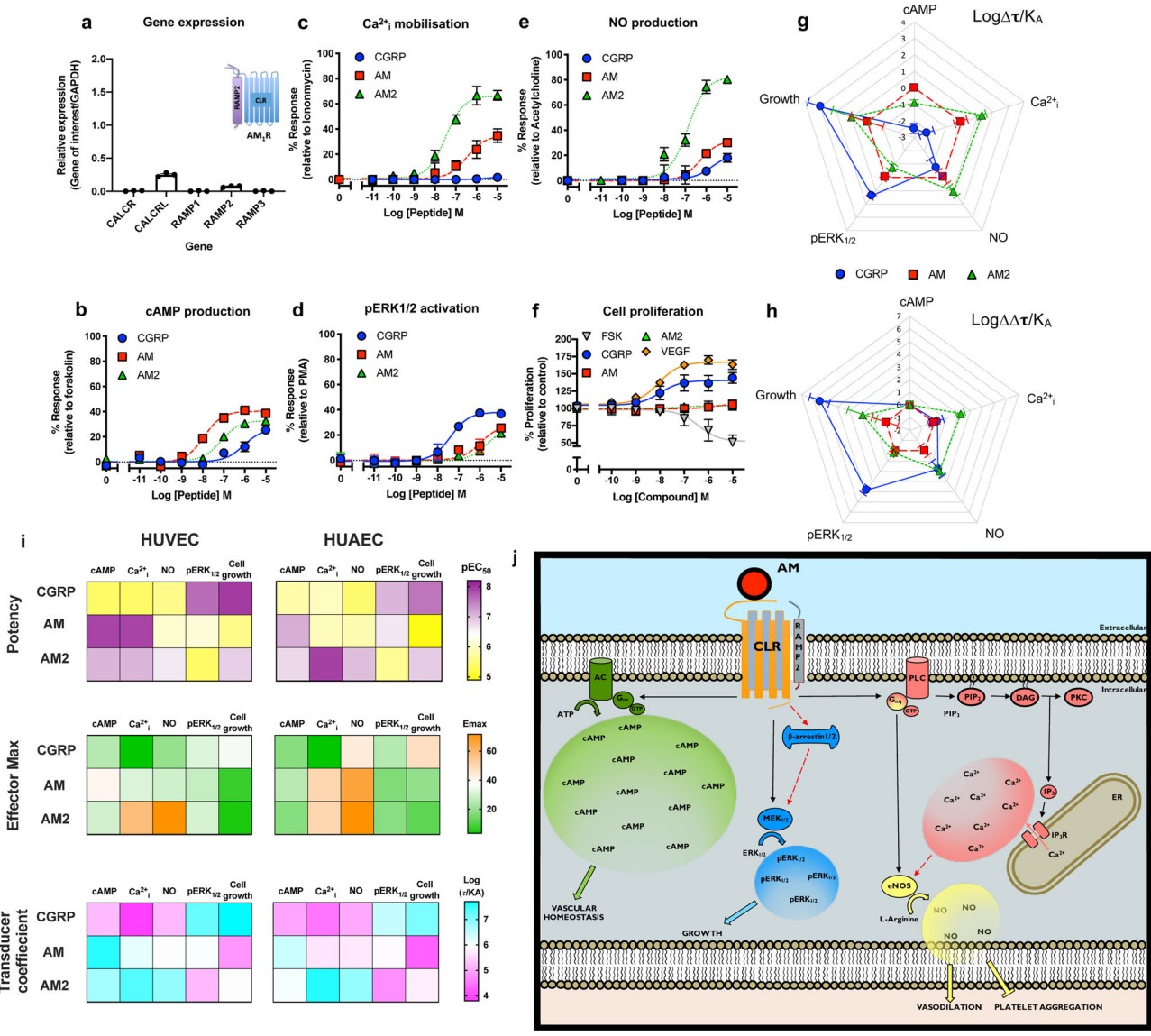

**Fig. 1 CGRP family peptide signalling bias in HUVECs. a** Expression of *CALCR, CALCRL, RAMP1, RAMP2*, and *RAMP3* genes in HUVECs. Data normalised to *GAPDH* expression. $n = 3$ independent experiments. **b–f** Dose–response curves were constructed for HUVECs stimulated with CGRP, AM or AM2 and the cAMP levels quantified relative to forskolin (100 μM) ($n = 7$) (**b**), mobilisation of $Ca^{2+}_i$ relative to ionomycin (10 μM) ($n = 6$) (**c**), intracellular $ERK_{1/2}$ phosphorylation relative to PMA (10 μM) ($n = 4$) (**d**), total nitric oxide production relative to acetylcholine (10 μM) ($n = 3$) (**e**), and extent of cell proliferation (after 72 h) relative to vector treated control and VEGF ($n = 3–6$) (**f**). Data are analysed using a three-parameter non-linear regression curve or the operational model of receptor agonism[27]. **g–h** Signalling bias plots were calculated as $\Delta Log(\tau/K_A)$ (**g**) or $\Delta\Delta Log(\tau/K_A)$ (**h**) for each agonist and for each signalling pathway. Determination of values requires normalisation to a reference agonist (AM) alone in (**g**), while for (**h**) values were normalised to both a reference agonist (AM) and a reference pathway (cAMP). **i** Heatmaps representing the signalling properties between HUVEC and HUAEC cells for potency, effector maximum and the transducer coefficient. All data represent mean ± SEM for *n* repeats. **j** Representation of the signalling outcomes as a result of AM-mediated receptor activation in a HUVEC. Solid arrows indicate known pathways. Dashed arrows represent possible pathways.

activate the canonical second messenger pathway to varying extents that is important, but their ability to differentially activate a multitude of downstream pathways. Having established that HUVECs express only one of the receptor-RAMP complexes responsive to our three peptides: CGRP, AM and AM2 with the expected potency, we next sought to quantify the extent of endogenous agonist-induced biased signalling through the AM1 receptor at other pathways. Consistent with these previous reports in recombinant systems[19] we were able to observe concentration-dependent increases in $Ca^{2+}_i$ in HUVECs upon application of AM and AM2 but little or none with CGRP (Fig. 1c, and Supplementary Table 1). In contrast to the stimulation of cAMP, AM2 was more potent than AM suggesting that a

non-cognate agonist can have a distinct and more potent effect than the cognate agonist at certain pathways endogenously. Importantly, all responses could be abolished with the co-treatment of the $G\alpha_{q/11/14}$ inhibitor YM-254890[24] (Supplementary Fig. 1i) thereby confirming CLR-based pleiotropy in primary endothelial cells.

We subsequently turned our attention to the extracellular signal-regulated kinase 1/2 ($ERK_{1/2}$) pathway (assayed after 5 min stimulation, when the response had reached a plateau (Supplementary Fig. 1j)) where we found that, again, the 'cognate' agonist (AM) was not the most potent. Perhaps surprisingly, CGRP (the agonist reported to be the least potent at cAMP production at the AM1 receptor) was the most potent at stimulating $ERK_{1/2}$

phosphorylation (Fig. 1d and Supplementary Table 1). Thus, despite this being designated an AM1 receptor, it is CGRP and not AM that produces physiologically relevant signalling via the $ERK_{1/2}$ pathway.

**Physiological consequences of CGRP-based peptide agonist bias in primary endothelial cells.** As we were exploring the AM1 receptor in its native environment, we sought to discover whether the distinct patterns of agonist bias we have observed with CGRP, AM and AM2 were reflected further downstream in physiologically relevant outputs. We considered two potential physiological outcomes with important therapeutic potential – the generation of NO (a vital modulator of vascular homeostasis) and cell proliferation. NO, generated through endothelial NO synthase in endothelial cells[25] promotes vasorelaxation/dilation[26]. In HUVECs we observed that all three agonists could evoke NO synthesis in the order of potencies AM2 > AM > CGRP (Fig. 1e and Supplementary Table 1) although both AM and CGRP were partial agonists for this pathway with the potencies closely resembling the trends observed for $Ca^{2+}_i$ mobilisation. Indeed, a direct correlation between $Ca^{2+}_i$ mobilisation and NO production in endothelial cells was confirmed through the application of YM-254890 which abolished all NO release (Supplementary Fig. 1k). Such observations are consistent with the role of increases of $Ca^{2+}_i$ concentrations leading to endothelial NO synthase stimulation[25] but to the best of our knowledge, these have not been demonstrated previously for AM2.

Beyond NO production we also measured the long-term cell proliferation (72 h) response to the three peptides in HUVECs. Here CGRP most potently promoted cell growth (Fig. 1f and Supplementary Table 1). This is consistent with the data we describe for phosphorylation of $ERK_{1/2}$ suggesting proliferation is not mediated via a cAMP-dependent pathway. This was further corroborated by the observation that application of the non-selective adenylyl cyclase activator forskolin induced a concentration-dependent inhibition of cell proliferation. Together, these data suggest that CLR exerts important cellular effects in a $G\alpha_s$-independent manner thus unveiling previously undocumented abilities for CGRP to promote proliferation in human cells through the AM1 receptor.

Whilst the differences in orders of potency seen with cAMP, $Ca^{2+}_i$, and $ERK_{1/2}$ provide strong evidence for bias, to formally confirm this and to remove potential confounding issue of system bias (which may arise due to the differential expression of signalling components or cofactors in the cellular background of choice) we fitted our HUVEC data with the operational model of receptor agonism[27] (Fig. 1g, h and Supplementary Table 1). This gives the transducer coefficient $Log(\tau/K_A)$; effectively the efficacy of an agonist to produce a given response normalised to its functional affinity. Agonist bias is calculated by computing $\Delta Log(\tau/K_A)$; the difference in transducer coefficient for each response compared to AM (Fig. 1h) and then the bias factor, $\Delta\Delta Log(\tau/K_A)$, where there is a second round of normalisation with respect to the cAMP pathway as well as AM (Fig. 1h). This analysis reinforced the notion that AM2 is biased towards $Ca^{2+}_i$ mobilisation and NO production while CGRP favours $pERK_{1/2}$ activation and cell proliferation.

We next wondered if the patterns of AM1 receptor bias applied to other endothelial cell lines. We performed the same panel of assays using human umbilical artery endothelial cells (HUAECs) which also solely express transcripts for *RAMP2* and *CALCRL* (Supplementary Fig. 2a–h, Supplementary Table 1). Indeed we were able to demonstrate a strong similarity in the signalling profiles between the two endothelial cells across the five different pathways (Fig. 1i and Supplementary Fig. 3a, b)

with significant correlations in potency (Supplementary Fig. 3c; $r = 0.73$–95% confidence interval 0.35–0.90; $p = 0.0019$) and the transducer coefficient (Fig. 1i, Supplementary Fig. 3c; $\tau/K_A$; $r = 0.94$–95% confidence interval 0.84 to 0.98; $p < 0.0001$ (to 4 decimal places)) suggesting primary endothelial cells share common AM1 receptor signalling properties (Fig. 1j).

**AM1 receptor-mediated cAMP accumulation and $pERK_{1/2}$ activation exemplify agonist bias.** The mechanism by which adenylyl cyclase is regulated involves competition between $G_s$ (activation) and members of the $G_{i/o}$ (inhibition) family of G proteins. Semi-quantitative RT-PCR in HUVECs and HUAECs revealed the presence of the same $G\alpha$ subunits (Supplementary Fig. 3d-e) and $\beta$-arrestins in the two cell lines. We and others have documented how the AM1 receptor (analogous to other class B1 GPCRs) couples to the inhibitory G proteins[19,28,29] although this is often observed in overexpression systems and is cell type dependent. Application of pertussis toxin (PTX), which ADP-ribosylates the inhibitory G proteins (except for $G_z$), to the HUVECs revealed a dose-dependent increase in cAMP accumulation (Fig. 2a) and suppression of $ERK_{1/2}$ phosphorylation upon application of CGRP and AM2 but not AM (Fig. 2b). These data, consistent with our previously reported work[19] suggests that only the non-cognate agonists (CGRP and AM2) can recruit $G_{i/o}$ proteins to the CLR, and, particularly in the case of CGRP, the purpose of this is to bias the response away from cAMP and towards other pathways such as $pERK_{1/2}$.

This did, however, pose the question as to how AM modulates the $pERK_{1/2}$ response? Inhibition of protein kinase A had no effect (Fig. 2c) however antagonism of $G_{q/11/14}$ signalling reduced the potency of AM-mediated $pERK_{1/2}$ activation (Fig. 2d). More strikingly, inhibition of the exchange proteins directly activated by cAMP 1/2 significantly attenuated both the potency ($p = 0.0078$) and magnitude of the maximal response (Fig. 2e). Taken together, these data highlight the wide array of different G protein couplings and their interlinking actions upon downstream signalling events for the AM1 receptor. These couplings have not been engineered so are not enhanced by overexpression artefacts and thereby represent pure endogenous agonist bias.

**RAMP isoform is essential for CLR-mediated agonist bias.** One of the advantages of using recombinant cell lines is the ability to switch the expressed GPCR or RAMP to observe effects on agonist bias. However, these recombinant systems do not allow for observations of physiological bias. Thus, we next sought to determine the effects of CGRP-based agonist bias in primary cells where the endogenous RAMP had been switched using gene deletion followed by lentiviral reintroduction in HUVEC cells.

We used lentiviral CRISPR-Cas9 to knockout the *RAMP2* gene from HUVECs using a pooled sgRNA strategy of using three sgRNAs in separate lentivirus (Supplementary Fig. 4a) which were selected using a puromycin resistance cassette (Supplementary Fig. 4b) to increase our efficiency of editing (95% as confirmed by Sanger sequencing (Supplementary Fig. 4c, d, Supplementary data 3, 4) and TIDE[30] analysis). We confirmed the loss of *RAMP2* by qRT-PCR (Supplementary Fig. 4e) although the expression of the $G\alpha$ subunits the $\beta$-arrestins remained consistent with wild type HUVECs (Supplementary Fig. 4f). The rate of proliferation of the cells was unchanged (Supplementary Fig. 4g) but as expected, there was no longer any functional response following stimulation with all three CGRP-based agonists (Supplementary Fig. 4h–l). We next introduced, using lentiviral overexpression and blasticidin selection, the open reading frame of *RAMP1* into our HUVECΔRAMP2 cell line to, in effect, switch the expressed GPCR from the AM1 receptor to

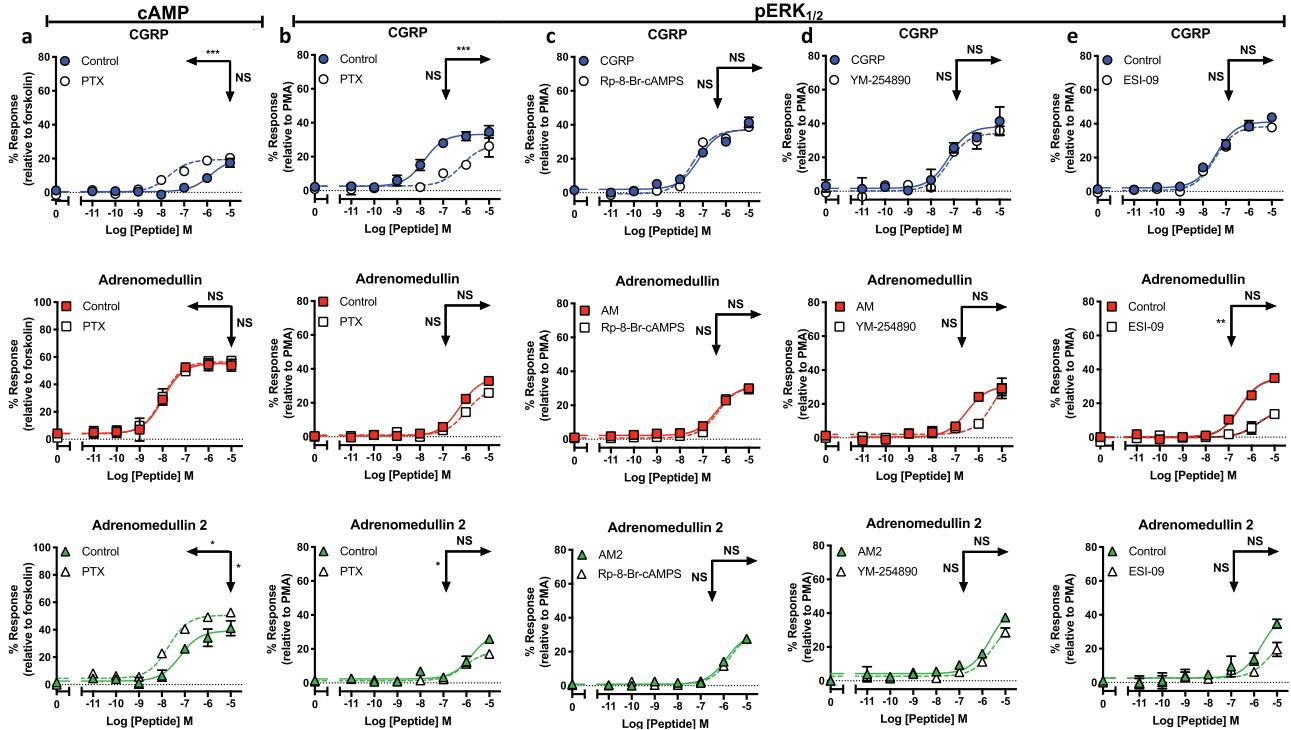

**Fig. 2 Non-cognate Gα couplings at the AM1 receptor complex modulates cAMP accumulation and ERK$_{1/2}$ phosphorylation. a** Characterisation of cAMP accumulation in response to stimulation by CGRP, AM, and AM2 with and without PTX treatment relative to forskolin (100 µM) ($n = 4$). **b–e** Characterisation of ERK$_{1/2}$ phosphorylation in response to stimulation by CGRP, AM, and AM2 with and without PTX treatment relative to PMA (10 µM) ($n = 4$) (**b**), with/without Rp-8-Br-cAMPS (10 µM) ($n = 3$) (**c**), with/without YM-254890 (100 nM) relative to PMA (10 µM) ($n = 3$) (**d**) and with/without ESI-09 (100 µM) ($n = 3$) (**e**). Data are analysed using a three-parameter non-linear regression curve. All data represent mean ± SEM of n independent experiments. Statistical significance determined compared to control using an unpaired Student's $t$ test with Welch's correction (*$p < 0.05$; **$p < 0.01$; ***$p < 0.001$). NS denotes no statistical significance observed. Rows show pEC$_{50}$, and vertical arrows show E$_{max}$ statistical significance.

the CGRP receptor. mRNA levels were quantified demonstrating successful introduction of a high level of *RAMP1* expression (Fig. 3a). We next performed cAMP accumulation assays using CGRP, AM and AM2 confirming that a functional CGRP receptor was formed in these modified HUVECs (Fig. 3b and Supplementary Table 2). Reassuringly, we now observed that CGRP was the most potent agonist for the stimulation of cAMP —as expected for a cell line expressing the CGRP receptor (CLR-RAMP1). Perhaps more interestingly, CGRP was also the most potent at mobilising Ca$^{2+}$$_i$ (Fig. 3c and Supplementary Table 2) and this was also the case in the associated NO production (Fig. 3d and Supplementary Table 2). Comparison of the ERK$_{1/2}$ phosphorylation (Fig. 3e and Supplementary Table 2) highlighted that AM was now the most potent agonist; a clear switch from wild type HUVEC cells where CGRP was the most potent. This followed to proliferation where AM was also the most potent ligand, although both AM2 and CGRP could also promote growth (Fig. 3f and Supplementary Table 2) and contrasted with the wild type HUVECs where neither could cause proliferation. Thus switching the RAMP in the HUVEC cell line appeared to have a dramatic effect on the agonist bias observed and the functional consequence (Fig. 3g, h and Supplementary Table 2)— beyond just cAMP accumulation as would be excepted. However, it should be noted that as *RAMP1* expression was high we should be cautious in our direct comparisons between the wild type HUVECs and our RAMP1-HUVEC cell line.

**Endogenous agonist bias at the CGRP receptor in primary human cardiac myocytes.** To provide a comparison for RAMP1-HUVEC signalling with a primary cell line we turned to primary human cardiomyocytes (HCMs) since these cells only expressed

CLR and RAMP1 (Fig. 4a); analogous to endothelial cells, HCMs do not express a functional calcitonin receptor (Fig. 4a and Supplementary Fig. 5a). To confirm that the mRNA expression translated to functional receptor expression we performed cAMP accumulation assays for the CGRP family of peptides (Fig. 4b and Supplementary Table 2). Here, CGRP was the most potent agonist followed by AM2 and AM, a pattern consistent with the expression of the CGRP receptor[20] (also confirmed by application of 100 nM olcegepant to inhibit cAMP accumulation for all three agonists while 100 nM AM22–52 had little effect (Supplementary Fig. 5b–g)). Although Gαi has previously been suggested to be important for PTX-sensitive effects from CLR[19], upon application of PTX to HCMs we were unable to observe any change in the potency or maximal signalling for any of the three peptide agonists (Supplementary Fig. 5h–j), perhaps because the transcript for *GNAi2* (Gα$_{i2}$) was lower than in the endothelial cells (Supplementary Fig. 5k). In contrast to the wild type HUVECs but analogous to the RAMP1-HUVECs cells, not only was CGRP able to stimulate G$_{q/11/14}$-mediated-Ca$^{2+}$$_i$ mobilisation in HCMs, but it was the most potent agonist (Fig. 4c and Supplementary Table 2). This did not directly translate to G$_{q/11/14}$-mediated-NO production (Fig. 4d, Supplementary Fig. 5l, m and Supplementary Table 2) since all three agonists generated responses that were less distinct from each other. When quantifying ERK$_{1/2}$ phosphorylation we observed that the cognate ligand (CGRP) was not the most potent (Fig. 4e and Supplementary Table 2). As in HUVECs, it was the least potent ligand at cAMP accumulation (AM) that was the most potent for ERK$_{1/2}$ phosphorylation so demonstrating that AM can produce functionally relevant signalling responses at the CGRP receptor. The order of potency for the three agonists for ERK$_{1/2}$ phosphorylation was replicated in

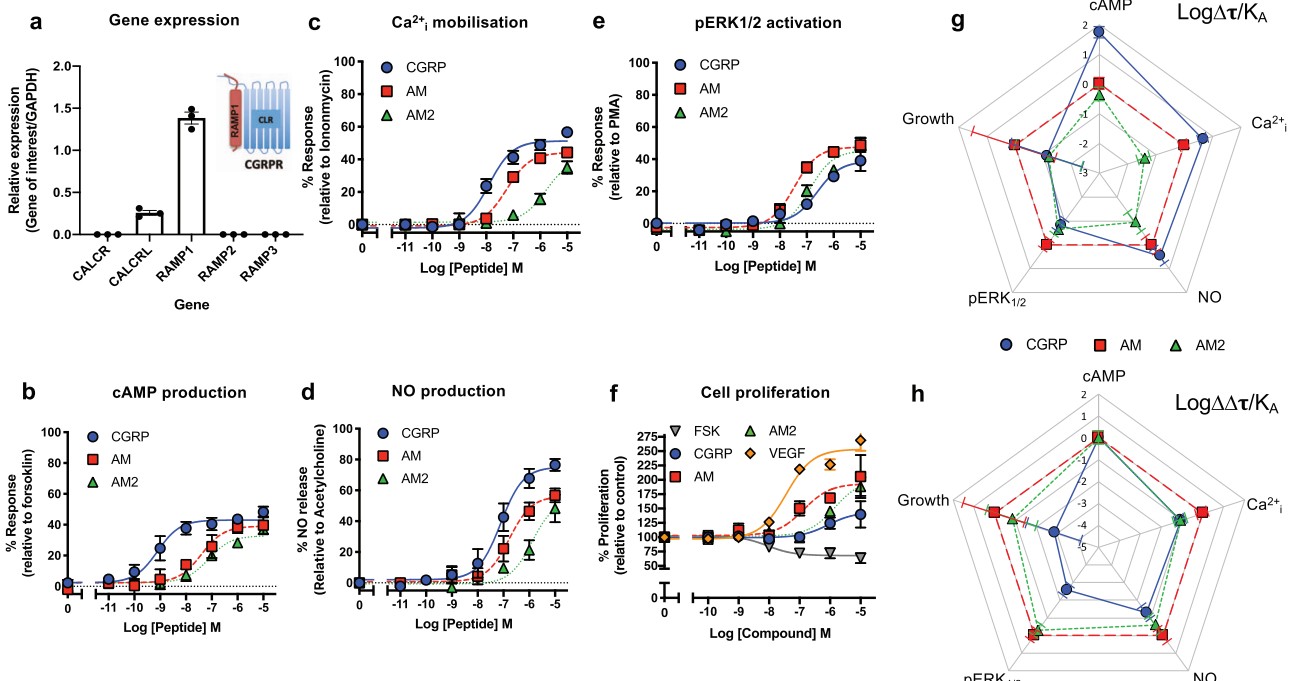

**Fig. 3 Switching RAMP expression in HUVECs produces unique signalling bias patterns for CGRP family of agonists. a** Expression of *CALCR, CALCRL, RAMP1, RAMP2,* and *RAMP3* genes in *RAMP1* expressing HUVECs. Data normalised to *GAPDH* expression. $n = 3$ independent experiments. **b–f** Dose–response curves were constructed for RAMP1 expressing HUVECs stimulated with CGRP, AM or AM2 and the cAMP levels quantified relative to forskolin (100 μM) ($n = 4$) (**b**), mobilisation of $Ca^{2+}_i$ relative to ionomycin (10 μM) ($n = 6$) (**c**), total NO production relative to acetylcholine (10 μM) ($n = 3$) (**d**), intracellular $ERK_{1/2}$ phosphorylation relative to PMA (10 μM) ($n = 3$) (**e**), and extent of cell proliferation (after 72 h) relative to vector treated control and VEGF ($n = 4$) (**f**). Data are analysed using a three-parameter non-linear regression curve or the operational model of receptor agonism[27]. **g–h** Signalling bias plots were calculated as $\Delta Log(\tau/K_A)$ (**g**) or $\Delta\Delta Log(\tau/K_A)$ (**h**) for each agonist and for each signalling pathway. Determination of values requires normalisation to a reference agonist (AM) alone in (**g**), while for (**h**) values were normalised to both a reference agonist (AM) and a reference pathway (cAMP). All data represent mean ± SEM for *n* repeats.

the long-term cell proliferation assays (Fig. 4f and Supplementary Table 2).

Analysis of the RAMP1-HUVEC signalling profile suggested a close overlap with the properties of HCMs for the five different signalling pathways as well as an opposing signalling profile to HUVECs (Fig. 5 and Supplementary Table 2). We confirmed this using correlation plots (Supplementary Fig. 6a–f) of both potency and transducer coefficient $Log(\tau/K_A)$, obtained from application of the operational model of receptor agonism[27] and in bias factors (Fig. 5a, b, Supplementary Table 2, 3). Overall, our results show that the pattern of bias seen with the CGRP receptor is robust and is transferred between the HUVEC and HCM cell line backgrounds.

## Discussion

We have shown that the CGRP family of endogenous peptides demonstrate biased agonism at the endogenous CLR in a physiological system; and that the RAMP expressed dictates the intracellular response and ultimately the physiological outcome. Many receptors have been shown to demonstrate agonist bias; but for the most part, this has been shown through synthetic ligands designed to target certain receptor pathways[2]. We have now shown that this is a process that can occur physiologically to direct different outcomes. Through elucidating distinct patterns of signalling bias that each peptide-receptor-RAMP produces, we have shown that bias is a naturally occurring phenomenon in a range of human cardiovascular cells. Furthermore, while we have only begun to scratch the surface of how important bias is physiologically, it is now clear it is an intrinsic part of endogenous CLR function, and we anticipate this is the case for many more

GPCRs that exhibit signalling bias in over-expression studies. We have also demonstrated the importance of studying GPCR second messenger signalling with the endogenous receptor in its native environment, with the distinct signalling patterns of AM2 we have uncovered providing a good example of this.

We have confirmed, using genome editing that, as anticipated by the co-expression models, that endogenous CLR is unable to function without RAMP expression. This provides, to the best of our knowledge, the first example of CRISPR-Cas9 interrogation of GPCR function in a primary cardiovascular cell. Furthermore, we have shown that the expression of a different RAMP in the HUVECs can switch the signalling bias of the CLR and associated peptide agonists, thus providing additional evidence that RAMP targeting could become a powerful therapeutic tool[31].

We have compared the pharmacology of these receptors in terms of cAMP accumulation to reports compiling multiple values from independent publications using the human receptor in transfected systems[22]. It was reassuring therefore to observe that CGRP, AM, and AM2 displayed similar trends in cAMP potency at CGRP/AM1 receptors in the primary cells to those seen in recombinant co-expression studies, as well as in the gene-edited RAMP1-HUVECs which reflected the cAMP data from transfected systems (Fig. 5c). In addition, we have performed a comprehensive analysis of the mechanisms used by AM1 receptor to stimulate $ERK_{1/2}$ phosphorylation in endothelial cells (Fig. 2). It is apparent that each agonist uses unique mechanisms to activate $ERK_{1/2}$. For both CGRP and AM2 it is mediated in a $G_{i/o}$-dependent manner, while AM uses a combination of $G_{q/11/14}$ signalling and exchange proteins directly activated by cAMP1/2 activation. None of the agonists appear to mediate

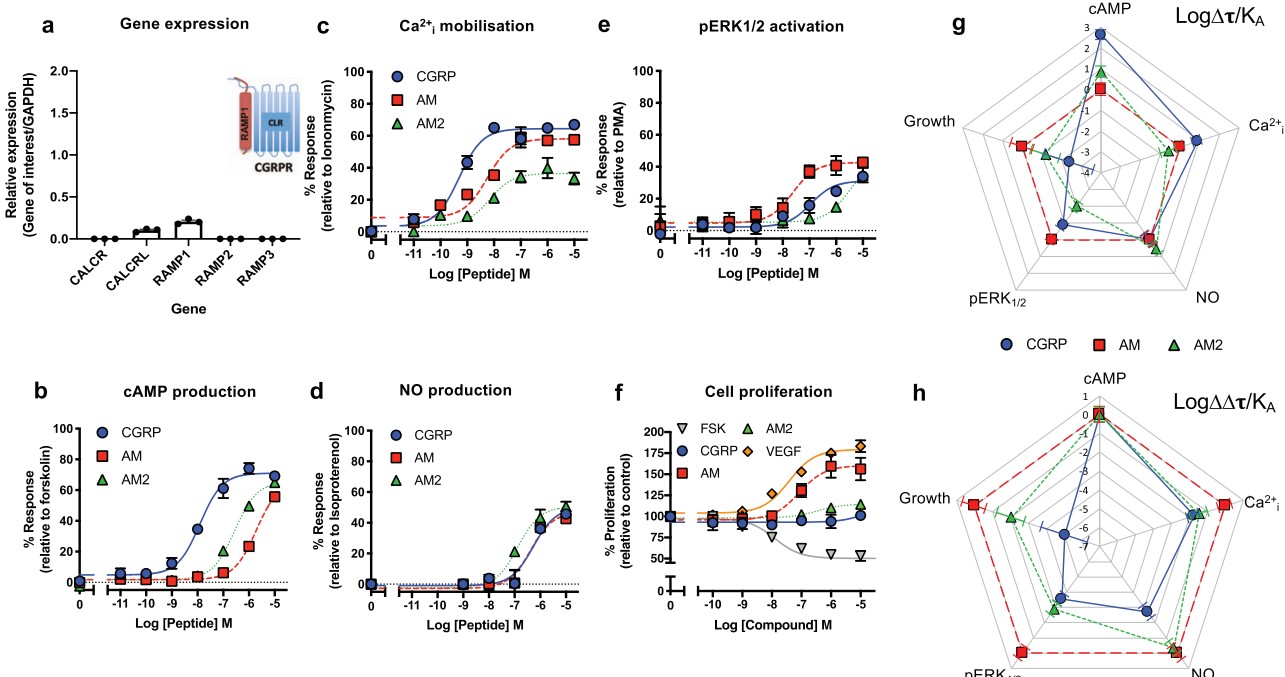

**Fig. 4 CGRP family peptide signalling bias in human cardiomyocytes. a** Expression of *CALCR, CALCRL, RAMP1, RAMP2,* and *RAMP3* genes in HCMs. Data normalised to *GAPDH* expression. $n = 3$ independent experiments. **b–f** Dose–response curves were constructed for HCMs stimulated with CGRP, AM or AM2 and the cAMP levels quantified relative to forskolin (100 μM) ($n = 6$) (**b**), mobilisation of $Ca^{2+}_i$ relative to ionomycin (10 μM) ($n = 4$) (**c**), total NO production relative to isoproterenol (10 μM) ($n = 3$) (**d**), intracellular $ERK_{1/2}$ phosphorylation relative to PMA (10 μM) ($n = 4$) (**e**) and extent of cell proliferation (after 72 h) relative to vector treated control and VEGF ($n = 5$) (**f**). Data are analysed using a three-parameter non-linear regression curve or the operational model of receptor agonism[27]. **g, h** Signalling bias plots were calculated as $\Delta Log(\tau/K_A)$ (**g**) or $\Delta\Delta Log(\tau/K_A)$ (**h**) for each agonist and for each signalling pathway. Determination of values requires normalisation to a reference agonist (AM) alone in (**g**), while for (**h**) values were normalised to both a reference agonist (AM) and a reference pathway (cAMP). All data represent mean ± SEM for *n* repeats.

their $ERK_{1/2}$ stimulation through the so-called cognate pathway, cAMP accumulation. The data presented is $ERK_{1/2}$ phosphorylation after 5 min and it will be of interest to determine the mechanisms and spatial locations that facilitate long term $ERK_{1/2}$ phosphorylation.

What has become clear in this study is that each of the endogenous ligands has very specific potencies at each pathway measured, whether it is at AM1 receptor in HUVECs/HUAECs or CGRP receptors in HCMs or RAMP1-HUVEC cells. Each peptide generates its own unique signalling profile (Fig. 5d–f). In a concentration-dependent manner, they individually recruit distinct G proteins in a manner regulated by the RAMP. This leads to a specific pattern of second messenger production and therefore a 'signalling barcode' for the cell to interpret and produce further physiologically necessary downstream responses. Expression analysis reveals in our three primary cell lines that each only expresses mRNA above the detection threshold for CLR and one RAMP. Combined with the cAMP signalling profile for each it appears that endothelial cells and HCM are excellent primary model cell types for the analysis of how the AM1/CGRP receptors signal in vivo.

It is worth noting that the present method of classifying receptors for CGRP and AM is based upon their potencies at cAMP production in addition to their affinities in binding assays[32]. This method arose due to the assumption that cAMP was the most physiologically relevant pathway. Here, we have demonstrated that different potencies are observed for agonists and these lead to physiologically relevant outcomes. As such we need to carefully consider how we classify CGRP-related receptors in the future and, more widely, all GPCRs that exhibit agonist bias.

We can also consider this work in the wider context of the organs and systems these cells are found in as this sheds light on some of the pathways, and involvement of bias in some of the established roles of CGRP family peptides in the cardiovasculature. It has long been recognised that all three peptides show pleiotropic signalling, activating multiple G proteins and signalling pathways[33] (and indeed this continues in current literature[34–36]), but it has previously not been possible to fit this into any framework. We suggest our current observations on RAMP-directed bias may assist with this.

AM has a multitude of important roles in vascular homeostasis;[6] one of which is regulating endothelial barrier function[37]. It is thought to cause barrier stabilisation and protect against infection mediated junctional protein disappearance, all brought about initially through cAMP production[38]. This is supported by our work demonstrating AM produces a potent cAMP response and is biased towards this pathway. It is also well documented that AM is a potent vasodilator known to mediate some of its vasodilatory effects through NO release from vascular endothelial cells[25,39] which we have pharmacologically profiled here.

In contrast, the precise role of AM2, which is also found in endothelial cells, has been unclear. We now provide evidence that AM2 is a potent stimulator of $Ca^{2+}_i$ mobilisation and NO synthesis. It is possible therefore that this plays a vital role at least in umbilical endothelial cell physiology, and indeed wider vascular physiology. Thus, the finding of AM2's greater potency than AM at eliciting NO release via $Ca^{2+}_i$ mobilisation may have great therapeutic potential.

Interestingly, in vascular endothelial cells CGPR inhibits adenylyl cyclase through $G_{i/o}$ and predominantly signals through $pERK_{1/2}$, and proliferation. The link between $pERK_{1/2}$ and

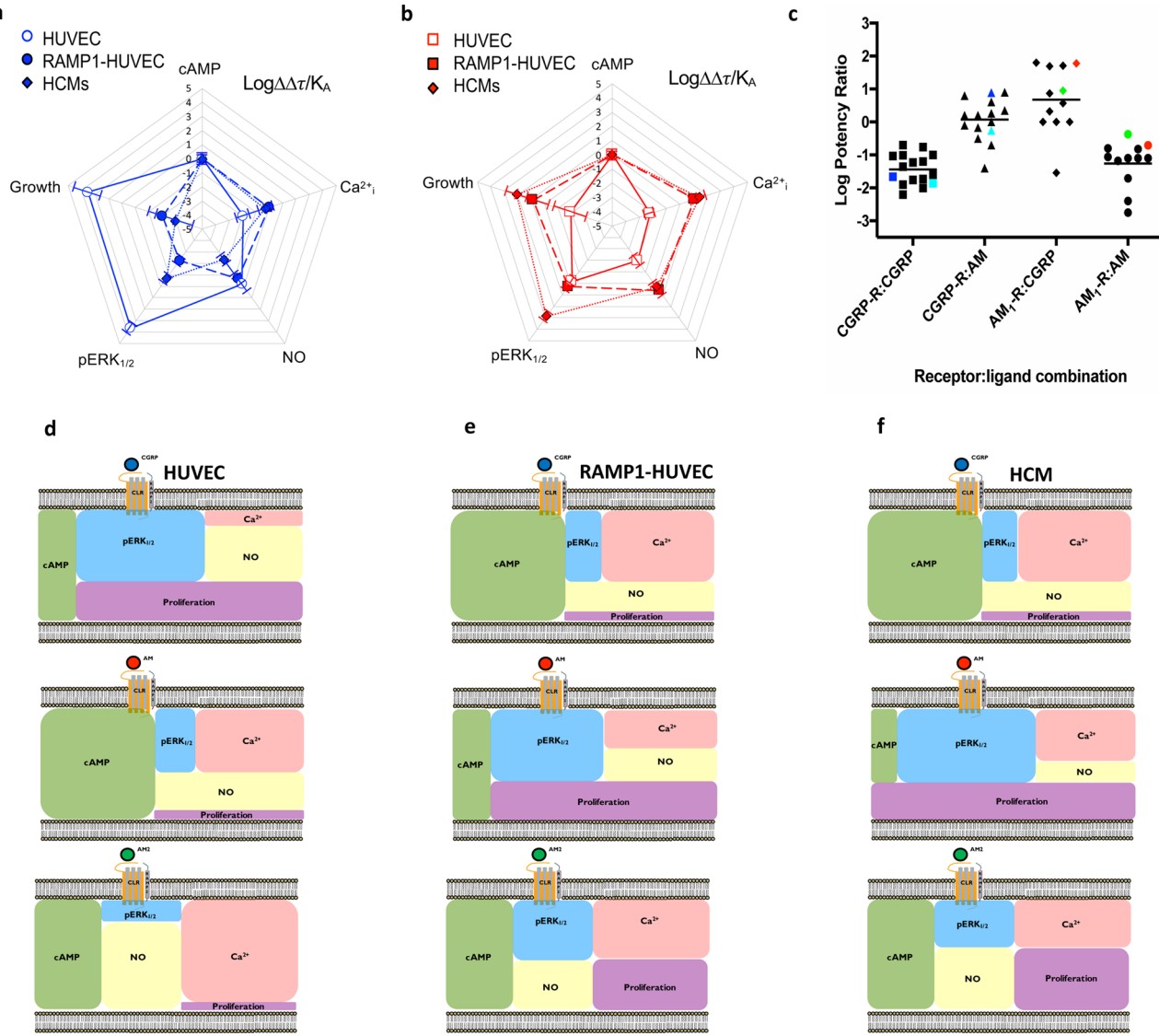

**Fig. 5 CGRP signalling bias in RAMP1 expressing HUVECs correlates with that in HCMs. a** Signalling bias plots were calculated as $\Delta\Delta Log(\tau/K_A)$ for CGRP in the three cell lines, HUVECs, *RAMP1* expressing HUVECs and HCMs for each pathway. Values have been normalised to a reference agonist (AM2) and the reference pathway (cAMP) for all three cell lines. **b** As for (**a**) except the calculated values are for AM. **c** Log potency ratios (as measured by the accumulation of cAMP) calculated as Log (EC$_{50}$AM2/EC$_{50}$ agonist). Data are compiled from[12,19]. HUVECs and HUAECs are shown in red and green respectively, HCMs in cyan and RAMP1-HUVECs in blue. **d–f** Schematic representation of the signalling bias produced by CGPR (**d**), AM (**e**) and AM2 (**f**), and the intracellular 'signalling codes' they bring about based on the potencies recorded at individual pathways in HUVECs, RAMP1-HUVECs, and HCMs.

cellular metabolism/proliferation is well established[40–42], as well as in endothelial cells specifically[43–45]. We have shown that where an agonist has biased signalling towards ERK$_{1/2}$ phosphorylation, this is carried through to long term cellular proliferation. Importantly, this shows overall that two non-cognate ligands, often not considered important for receptor function, do in fact have important signalling and physiological roles/capabilities. In addition, we have demonstrated that endogenous pERK$_{1/2}$ can come from a variety of sources depending on the stimulating ligand. Together this shows that CLR initiates a multitude of intracellular pathways beyond simply G$_s$ and cAMP/protein kinase A in physiologically relevant cells. Therefore, our data adds further evidence that AM for AM1 receptor and CGRP for CGRP receptor should only be considered the cognate ligands in terms of G$_s$-mediated cAMP signalling, and when looking at the physiology of RAMP2 in endothelial cells and the vasculature as a whole, CGRP and AM2 should be considered alongside AM for their different and potentially complementary roles.

In the heart, there are multiple reports that CGRP has a cAMP-mediated positive inotropic and chronotropic effect[5,46,47], and our data showing its strong response in cAMP accumulation assays on human myocytes (combined with its overall bias towards this pathway) supports this. There are contrasting reports in the literature over AM's effect on heart contractility;[7,48] with some suggestions that it is a positive inotrope acting in the same cAMP driven manner as β-adrenoceptor agonists[49], while others report it having negative inotropic effects[50,51]. Here we show that AM promotes a cAMP response through CGRP receptor in human cardiomyocytes, but it has weak potency. This may provide some context/explanation for the contradictory literature reports. Furthermore, our report has clearly revealed that cAMP is not AM's primary signalling pathway in HCMs and that it is biased towards pERK$_{1/2}$ and cell proliferation rather than cAMP and positive inotropy. Nevertheless, evidence suggests that AM has an important role in the human heart. This includes the observed elevation of AM in the failing heart[52]. Here, we have

utilised proliferating human ventricular myocytes in vitro and shown that AM (but not CGRP or AM2) exhibits signalling bias specifically towards $pERK_{1/2}$ and enhancing proliferation in these cells. This work highlights AM as a peptide hormone that may promote cardiac regeneration naturally in vivo and provides a cellular mechanism for this. This may also explain the elevation of AM in heart failure[6] and the clinical trial data showing that AM administration reduces infarct size[53]. For AM2, its effect on contraction of the heart is blocked by both inhibitors of both protein kinase A and C, suggesting multiple signalling pathways are activated by this peptide in-vivo[54]. This pleiotropy in signalling pathways is also present in vascular cells, which are themselves far from homogeneous. This may be relevant to the heterogeneity observed in the actions of CGRP and for example, the role of NO or other signalling pathways in vasodilation. Indeed, CGRP has been suggested, in earlier studies, to mediate endothelium-dependent relaxation involving NO via an unknown endothelial receptor[55]. We suggest the AM1 receptor would be a viable candidate for this 'unknown' receptor. It is also worth noting that changes in RAMP expression have been observed due to pathological changes in blood pressure and in response to drug treatment. These will all have complex effects as a result of bias and depending on the expression of CGRP, AM and AM2[56].

It should be noted that not all studies reveal pleiotropic signalling for the CGRP family of peptides[57]. We suggest that there is cell membrane (e.g. lipid composition) and cell line-specific factors (e.g. expression of G proteins) that influence the observed bias. It is also important to consider the temporal aspects of signal bias; in this study we have generally measured responses at what seems to be their peak up to 30-min stimulations. However, different patterns of bias may be observed if other time points are selected, particularly where signalling from internalised receptors is important.

In summary, we have gone beyond previous studies in recombinant systems to observe agonist bias. While we have focused upon CLR-RAMP complexes, our ability to switch the RAMP means we are, in effect, switching the expressed receptor and, therefore our work, has general applicability to all GPCRs. Our data highlights how endogenous agonist bias can have profound consequences for the cell and how important cell background is in regulating this process. This work may even go as far as to suggest that to fully understand bias at a GPCR, it must be considered in its native environment. While our work takes an important step closer to understanding how the CGRP family of peptides and receptors function on a cellular level in the human cardiovascular system, it also highlights the importance of endogenous agonist bias as a concept and emphasises its long-term consequences for drug design.

## Materials and methods

**Cell culture**. HUVECs and HUAECs (were both sourced from PromoCell, Germany; C-12250 and C-12252 respectfully) were cultured in Endothelial Cell Growth Media (ECGM) (PromoCell). Human Cardiac Myocytes (PromoCell, C-12811) were grown in Cardiac Myocyte Growth Media (CMGM) (PromoCell). All cell lines were cultured in media containing 10% heat inactivated Foetal Bovine Serum (FBS) (Sigma, USA). Cells were grown in 25 $cm^2$ flasks or 75 $cm^2$ flasks depending on cell density required. They were passaged approximately every 4 days depending on confluency with a final volume of 10 ml produced from 1 ml of the previous cell culture and 9 ml of the growth medium in 75 $cm^2$ flasks, or 1 ml using 4 ml in 25 $cm^2$ flasks, and used from passage 2–6 (with the exception of HUVECΔRAMP2 and RAMP1-HUVEC). All cells were grown with 1% antibiotic antimycotic solution (100x Sigma, USA). The cells were maintained in an incubator (37 °C, humidified 95% air, 5% $CO_2$) between passaging.

**Genome engineering**. HUVECs with the *RAMP2* gene knocked out were generated by CRISPR/Cas9 homology-directed repair as described previously[58]. We chose the HUVEC cell line with which to perform the gene editing since we have been able to grow HUVECs beyond passage 6 to passage 14 before loss of CGRP based signalling responses are observed (Supplementary Fig. 7), and it is necessary

to grow them past passage 6 to develop the RAMP2 null cells. We used a pooled sgRNA strategy using three sgRNAs in separate lentivirus (Supplementary Fig. 4a) which were selected using a puromycin resistance cassette (Supplementary Fig. 4b) to increase our efficiency of editing. The sgRNA sequences were designed (5′-CGC TCCGGGTGGAGCGCGCCGG-3′), (5′-TCCGGGTGGAGCGCGCCGGCGG-3′), and (5′-CCCGCGTCTCCCTAGGACCCGA-3′) for Cas9 targeting to the human *RAMP2* gene (Sigma, US). All guides were delivered in the LV01 vector (U6-gRNA: ef1a-puro-2A-Cas9-2A-tGFP) vector provided by (Sigma-Aldrich, US). Sequences were verified by Sanger sequencing. The control cell line was established by transduction of LV01 vector not containing sgRNA targeted to *RAMP2* gene. HUVEC cells were seeded in 6 well plates at a cell density of 160,000 cells/well and maintained at 37 °C in 5% $CO_2$ with Complete Endothelial Cell Growth Media containing 100 μg/ml streptomycin (Sigma-Aldrich, US). 24 h after seeding virus containing individual sgRNA/Cas9 constructs were pooled and transduced into cells at a high multiplicity of infection of 10, ensuring that each cell is infected by several lentivirus and increasing the likelihood of achieving knockout. Transduction was performed in media containing 8 μg/ml Polybrene (Sigma, USA). Cells were cultured for 24 h then treated with Puromycin (1 μg/ml) (Thermo Fisher Scientific, UK) for 3 days to select for transduced cells. Cells then cultured without puromycin and expanded before cells were collected for genotyping by Sanger sequencing, qRT-PCR, and functional assays. All data shown were from cells expanded from these colonies. *RAMP1* expression achieved through transduction of virus containing RAMP1 MISSION TRC3 ORF plasmid (pLX_304) (Sigma, US) into HUVEC-ΔRAMP2. HUVEC cells were seeded in 6 well plates at a cell density of 160,000 cells/well and maintained at 37 °C in 5% $CO_2$ with Complete Endothelial Cell Growth Media containing 100 μg/ml streptomycin (Sigma, US). 24 h after seeding, virus containing the ORF construct was transduced into cells in media containing 8 μg/ml Polybrene. Cells were cultured for 24 h then treated with blasticidin (5 μg/ml) (Thermo Fisher Scientific, UK) for 6 days to select for transduced cells. Cells were collected for genotyping by qRT-PCR and expanded for functional assays. All 'HUVEC-RAMP1' data shown were from cells expanded from these colonies.

**Immunofluorescence**. HUVEC cells were seeded in Cell Carrier Ultra 96-well plate (PerkinElmer, Boston, MA, US) at a cell density of 160,000 cells/well and maintained at 37 °C in 5% $CO_2$ with Complete Endothelial Cell Growth Media containing 100 μg/ml streptomycin. Cells were washed twice with PBS, fixed with 4% paraformaldehyde in PBS (10 min, room temperature) then washed three times with PBS. The cells were permeabilized with 0.05% Tween 20 in PBS (60 min, room temperature), and then incubated in 10% goat serum in PBS (60 min, room temperature). The cells were then incubated in primary antibody for Cas9 protein (Cell signalling technology, MA, US) 7A9-3A3, diluted 1/700 in PBS/0.05% Tween/ 3% BSA at 4 °C, overnight and protected from light. The cells were washed three times with PBS and incubated with AlexaFluor 488 goat anti-mouse (Invitrogen A11001, 1/500) (1 h, room temperature) protected from light. Cells were washed three times with PBS, then nuclei were stained with Hoechst (Invitrogen) (1/2000 in PBS, 10 min, room temperature). Cells were then washed three times with PBS w/o $Mg^{2+}$ or $Ca^{2+}_i$ and imaged at 20x magnification (Cell Voyager 7000 S, Yokogawa).

**Sequencing of genomic loci**. Genomic DNA was extracted from virally transduced HUVEC cells by: collecting approximately 10,000 cells, washing in PBS (sigma-Aldrich, US) and then lysing with DirectPCR Lysis Reagent (Viagen Biotech, US) containing Proteinase K (Qiagen, Germany) at 0.4 mg/ml. The lysate was incubated at 55 °C for 4 h; 85 °C, for 10 min; 12 °C for 12 h. PCR reaction was then set up in (20 μl) as follows: 2x Flash Phusion PCR Master Mix (Thermo Fisher, US) (20 μl), forward primer (5′-AATTCGGGGAGCGATCCTG -3′) (Eurogentec, Belgium) (1 μl)(10μm), reverse primer (5′- GAGACCCTCCGAAAATAGGC -3′) (Eurogentec, Belgium) (1 μl)(10μm), DNA (100 ng/μl)(1 μl), ddH2O (7 μl). The product was amplified by PCR using the following programme: 98 °C, 1 min; 35 x (98 °C, 10 s; 55 °C, 10 s; 72 °C, 15 s), 72 °C, 1 min; 4 °C, hold. PCR clean-up was performed prior to sequencing using the Illustra GFX PCR DNA and Gel-band Purification Kit (Illustra, Germany) according to the manufacturer's instructions. Editing of *RAMP2* gene was confirmed by Sanger sequencing (Eurofins–Supplementary Fig. 4c, d 1, Supplementary data 3, 4) and TIDE analysis[30].

**Quantitative real-time reverse transcription polymerase chain reaction (qRT-PCR)**. HUVECs were cultured as above in Complete Endothelial Cell Growth medium and plated in a 24-well plate at 100,000 cells/well. Media was then removed, and cells were washed in PBS (Sigma, UK). RNA was extracted and genomic DNA eliminated using an RNA extraction kit (Qiagen, Germany) as per the manufacturer's instructions. The yield and quality of RNA were assessed by measuring absorbance at 260 and 280 nm (Nanodrop ND-1000 Spectrophotometer, NanoDrop technologies LLC, Wilmington DE USA). RNA was used immediately for the preparation of cDNA using the Multiscribe reverse transcriptase. For the preparation of cDNA 100 ng of RNA was reverse transcribed using Taq-man reverse transcription kit (Life Technology, MA, USA) according to the manufacturer's instructions. Reactions were performed on a thermal Cycler as following: 25 °C, 10 min; 48 °C, 30 min; 95 °C, 5 min. cDNA was stored at −20 °C.

For each independent sample, qPCR was performed using TaqMan Gene Expression assays according to the manufacturer's instructions (Life Technologies, MA, USA) for *GAPDH, CALCR, CALCRL, RAMP1, RAMP2, RAMP3*, all Gα subunits and β-arrestins were plated onto fast microAmp plates containing 2 μl cDNA, 1 μl Taq-man probe, 10 μl Taq-man fast universal master mix (Applied Biosystems) and 10 μl ddH2O. Oligonucleotides used were as described in Weston et al.[19] and Routledge et al.[59]. For β-arrestin1 forward primer (5′-AAAGGGAC CCGAGTGTTCAAG-3′) (Thermo Fisher, US), reverse primer (5′–CGTCACA TAGACTCTCCGCT-3′) and β-arrestin2 forward primer (5′–TCCATGCTCC GTCACACTG-3′), reverse primer (5′–ACAGAAGGCTCGAATCTCAAAG-3′). PCR reactions were performed on ABI 7900 HT real time PCR system (Thermo Fisher Scientific, UK). The programme involved the following stages: 50 °C, 2 min; 95 °C, 10 min, the fluorescence detection over the course of 40x (95 °C, 15 s; 60 °C, 1 min). Data are expressed as relative expression of the gene of interest to the reference gene *GAPDH* where: Relative expression = 2-((Cq of gene of interest) − (Cq of *GAPDH*)).

### In vivo assays

*Measurement of intracellular cAMP*. All primary cell lines were cultured as above. On the day of the experiment media was removed and cells washed with PBS, before being dissociated with Trypsin-EDTA 0.05% (Gibco, UK) and then resuspended in PBS/BSA (0.1%) (Sigma, UK). Cells were immediately plated for use in cAMP assay as per the manufacturer's instructions, reagents used were provided by the LANCE® cAMP detection assay kit (PerkinElmer, Boston, MA, USA), in 384 well optiplates (PerkinElmer (Boston, MA, USA)) at 2000 cells/well in 5 μl aliquots[19]. Human (h) αCGRP, hAM and hAM2 (Bachem, Switzerland) were diluted in PBS/BSA (0.1%) with 250 μM 3-Isobutyl-1-methylxanthine (Sigma, UK), and used from 10 pM to 10 μM. Cells were incubated with compound for 30 min prior to adding detection buffer[28,60]. Plates were incubated for a further 60 min (room temperature) and then read on a plate reader (Mithras LB 940 microplate reader (Berthold technologies, Germany)). All responses were normalised to 100μM forskolin (Tocris, UK). Antagonist studies were performed in the same way through co-stimulation of the relevant concentration. Alongside control-treated cell. Experiments with PTX (Sigma, UK) required pretreatment (16 h) prior to assays.

*Measurement of Phospho-ERK1/2 (Thr202/Tyr204)*. Primary cells were grown in 6 well plates, on the day of the experiment media was replaced with serum free media 4 h prior to cell harvesting. Tyrpsin-EDTA was used to dissociate the cells and they were collected by centrifugation, counted and re-suspended HBSS/BSA (0.1%). Ligands were also diluted in HBSS/BSA. Cells were then plated on 384 well plates in 8 μl aliquots at a density of 20,000 cells/well. Next, ligands were added (4 μl) for 5 min stimulation at room temperature. Cells were then lysed as per to the manufacturer's instructions with 4 μl of lysis buffer (Cisbio phosphor-ERK1/2 cellular assay kit, Invitrogen, UK) for 30 min shaking at room temperature. The 2 specific antibodies; were pre-mixed in a 1:1 ratio. 4 μl of this was added to each well and the plate incubated for a further 2 h. Then fluorescence emissions were read at 665 nm and 620 nm using a Mithras LB940 microplate reader. Antagonist studies were performed in the same way through co-stimulation with PTX (200 ng/ml), Rp-8-Br-cAMPS (100 μM), YM-254890 (100 nM), or ESI-09 (100 μM) as appropriate alongside control-treated cell.

*Measurement of Intracellular Calcium mobilisation*. All cell lines were plated at 20,000 cells/well on 96-well black clear-bottom plates (Costar, UK) 24 h before the experiment. Media was removed, and cells were washed with Hank's Balance Salt Solution (HBSS) (Lonza, Switzerland) before cells were loaded with 10 μM Fluo-4/AM (Invitrogen, US) in the dark at room temperature for 30 min. Cells were then washed twice with calcium-free HBSS, then were left in 100 μl calcium-free HBSS for the duration of the assay. In conditions where Gαq/11/14 signalling is inhibited, cells were pre-treated with 100 nM YM-254890 (Alpha Laboratories, UK) (30 min)[22]. All assays were performed using the BD Pathway 855 Bioimaging Systems (BD Biosciences, UK), which dispenses ligands (20 μl) and reads immediately for 2 min. Data was normalised to the response seen with 10 μM Ionomycin.

*Measurement of cell proliferation*. Both endothelial cell lines and HCMs were seeded at a density of 2500 cells/well in a clear flat bottom 96-well plate (Corning, UK) and incubated at 37 °C in 5% CO2. After 24 h, cells were exposed to test compounds or vehicle, in complete endothelial cell growth media (HUVECs) or myocyte growth media (HCMs). Cells were incubated for a further 72 h at 37 °C in 5% CO2. After 72 h incubation, 5 μl of Cell Counting Kit–8 (CCK-8, Sigma, UK) was added to each well and cells were then incubated for another 2 h at 37 °C in 5% CO2 and in the dark[61]. The absorbance of each well was measured using a Mithras LB940 microplate reader with an excitation of 450 nm. The absorbance is directly proportional to the number of viable cells. Cell proliferation was calculated as a percentage of number of cells treated with vehicle alone.

*Measurement of nitric oxide production*. Endothelial cells and HCMs were cultured as above. 24 h prior to assay cells were plated on Costar 96-well black clear bottom plates at 40,000 cells/well. The assay was performed according to the

manufacturer's protocol. Briefly, cells pre-incubated with NO dye and assay buffer 1 (Fluorometric Nitric Oxide Assay Kit, Abcam, UK) for 30 min at 37 °C in 5% CO2. Any inhibitors requiring 30 min pre-treatment (YM/L-NAME/DMSO control) were added at this point. Ligand stimulation occurred immediately after this for 15 min at 37 °C in 5% CO2. Stain and ligand solution was removed, assay buffer II was added, and wells were read immediately. The absorbance was measured using a Tecan T200 (Thermo Fisher Scientific UK) reader with an excitation/emission of 540/590 nm. Endothelial cell responses were normalised to 10 μM acetylcholine[62]. HCM responses were normalised to 10 μM isoproterenol[63,64].

**Statistics and reproducibility**. All sample sizes were determined, and data analysed, in accordance with the guidelines described by Curtis M et al.[65]. All experiments were appropriately controlled using 'system pathway' agonists. If any of these pathway controls generated inappropriate responses, then the entire data set was removed from analysis. Tolerance for variation was <3-fold changes in potency for the system parameters tested. Data analysis for cAMP accumulation, Ca²⁺ᵢ mobilisation, NO accumulation, pERK1/2 activation, and cell proliferation assays were performed in GraphPad Prism 8.4 (GraphPad Software, San Diego). Data were fitted to obtain concentration–response curves using either the three-parameter logistic equation using to obtain values of Emax and pEC50 or the operational model of agonism[27]. Statistical differences were analysed using one-way ANOVA followed by Dunnett's post-hoc (for comparisons amongst more than two groups) or unpaired Student's t test with Welch's correction (for comparison between two groups). To account for the day-to-day variation experienced from the cultured cells, we used the maximal level of cAMP accumulation from cells in response to 100 μM forskolin stimulation was used as a reference, 10 μM ionomycin for Ca²⁺ᵢ assays, 10 μM phorbol 12-myristate 13-acetate (PMA) for pERK1/2 activation, 10 μM acetylcholine for NO production and 10 μM VEGF for cell proliferation. Emax values from these curves are reported as a percentage of these controls, and all statistical analysis has been performed on these data. Where appropriate the operational model for receptor agonism[27] was used to obtain efficacy (τ) and equilibrium disassociation constant (KA) values. In both cases, this normalisation removes the variation due to differences in days but retains the variance for control values. The means of individual experiments were combined to generate the curves shown. Having obtained values for τ and KA these were then used to quantify signalling bias as the change in Log(τ/KA)[19,28]. Error for this composite measure was propagated by applying Eq. 1.

$$Pooled\ SEM = \sqrt{(SEM_A)^2 + (SEM_B)^2} \qquad (1)$$

Where, $SEM_A$ and $SEM_B$ are the standard error of measurement A and B.

Correlations between pEC50 values or transducer coefficients Log (τ/KA) were assessed by scatter plot and Pearson's correlation coefficient (r) was calculated with 95% confidence interval.

**Reporting Summary**. Further information on research design is available in the Nature Research Reporting Summary linked to this article.

## Data availability

Sequencing data is available in Supplementary Data 3, 4. Source data for the graphs and charts in the main figures are given as Supplementary Data 1, 2 and 5–7. Any remaining information including the primary data for all pharmacological investigations is available from the corresponding authors on reasonable request.

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

## Acknowledgements

This work was supported by the Biotechnology and Biological Sciences Research Council [grant number BB/M000176/2] awarded to GL and DRP, a Medical Research Council Confidence in Concept award to GL and MH (MC_PC_17156) and the Endowment Fund for education from Ministry of Finance Republic of Indonesia (DS). AJC is funded through an AstraZeneca Scholarship.

## Author contributions

D.R.P., A.M., M.W. and G.L. conceived and designed the research; A.J.C., T.V. and D.S. performed the experiments; A.J.C., N.M. and D.G. designed the CRISPR-Cas9 experiments, A.J.C., G.L., M.H. and M.W. analysed data; A.J.C., M.H. and G.L. wrote the paper, D.R.P., A.M., M.W. and D.G. revised and edited the paper.

## Competing interests

The authors M.W., N.M. and D.G. declare the following competing interest: they are employees of, and shareholders in, AstraZeneca. The remaining authors declare no competing interests.
