## [Peer Review File · Communications Biology]

Reviewers' comments:

Reviewer #1 (Remarks to the Author):

This manuscript investigates receptor signalling for the calcitonin gene-related peptide (CGRP) signalling pathway involving the CGRP receptor and those related to it. The study involving primary human cardiovascular cells indicates the ability to influence function and further signalling via biased agonist activity. Should the title also include AM, as this is an important component of the manuscript?

Generally I found navigating around this manuscript, which contains highly relevant and novel data challenging. This is a problem throughout the manuscript.

The abstract can be improved upon, in that the novelty of the work is not coming through.

Could the authors add a schematic to enable the potential signalling pathways to be more easily understood at the introduction level?

Results.

1. Can the authors justify the use of human umbilical cells. This is not a typical vascular cell for a number of reasons; although I know later they support the data with cardiomyocytes. Is the result shown in Fig 1/S1 typical for microvascular/aortic sources of human cells from more typical sources too? Note there is a typo in Fig s1 legend line 2 which confuses. Note that not all figures are referred to in Fig 1, before jumping to Fig 2, which makes for a challenge to the reader.

2. If Fig S2D is important, why isn't it in the main manuscript?

In Fig S2 many of the graphs do not have a letter code. Also is S2E not important for sense of manuscript too. Why is it not in main manuscript?

Can the section on 'system bias' be more fully explained?

The exchange of RAMP2 for RAMP1 is clearly documents and nice work. I am slightly confused by the high level of RAMP1. Was it not possible to titrate the transfection level? I note that similar concentrations of the CGRP antagonist and the AM antagonist were used, but are they equally effective as antagonists?

Cardiomyocytes- This data is clear, but the results section ends abruptly, a further line of explanation would be useful.

By the end of the manuscript I am impressed by the high quality of the research and potential impact. However I believe the authors should attempt to explain it more easily and in a more straight forward manner. I think if anything the manuscript is trying to say too much, not helped by the complex nature of this receptor family. Could Fig 5I-L be utilised as a separate Fig for explanation in the discussion.

I am left wondering about the historical aspects to this study and how this bears on bias agonism. For example, there are a range of studies where various RAMPs have been shown to be raised in certain conditions, for example RAMP1 leading to a supposed upregulation of the CGRP receptor. Additionally, there are historical suggestions that the nature of the receptors may change and additionally that CGRP may be an NO-dependent vasodilator in some vascular tissues, but not others. Could the findings be put into better context with the historical findings?

Line 17 page 3 typo these/there

Reviewer #2 (Remarks to the Author):

In this work, the authors explore the ability of ligands to activate the CLR receptor in primary endothelial cells, with detailed insights into signaling pathways and the influence of co-expressed RAMPs. This represents an extension of insights gleaned from cells engineered to express recombinant receptor and RAMPs by focusing on natural cells and on naturally-occurring agonist ligands. This group of receptors and its ligands has already been demonstrated to be significant for the treatment of migraine and has potential importance for cardiovascular disease. Traditionally, the complexity of such a system with multiple receptors, RAMPs, and ligands can be most clearly elucidated taking a reductionist approach in such engineered cell systems, where each variable can be systematically adjusted and the impact determined. By working with primary cells *ex vivo*, the concepts developed previously can be tested in isolated cells. It would be ideal to extend this further to natural tissues and to *in vivo* observations.

Included in this report are two populations of endothelial cells (HUVECs and HUAECs), both of which are characterized to express only CLR and not CTR, and RAMP2 and not other RAMPs (AM1 receptor). Also included for limited observations are primary cardiac myocytes that express only CLR and RAMP1 (CGRP receptor). Each of the cells was carefully evaluated for expression of the relevant genes and for relevant signaling pathways with appropriate positive controls included.

Rank order potency of the natural agonists of the CLR-RAMP2 receptor (AM1 receptor) was as expected, when quantifying cAMP, as has been the tradition. Also, as expected, the signaling responses to these ligands were pleiotropic. The different patterns of signaling events for different ligands, viewed as "bias", was also clearly demonstrated. In this more natural setting than the engineered cell lines, the varied signaling events for different ligands was nicely demonstrated. It was particularly notable that AM2 was actually more potent than AM at this receptor, but this was dependent on calcium signaling after coupling with Gq/11.

Another interesting line of study was exploration further distally in signaling pathways, including NO and ERK pathways. The evaluation of different agonist effects in these cells provided new insights into what might be possible in mediating cell growth. The case was made that this is likely distinct from the classical focus of this receptor on AM and cAMP.

The authors went further than to only describe these events in the primary endothelial cells, by engineering a change in the RAMP expressed in those cells using gene editing. They showed that the intermediate product lacking RAMP2 exhibited no signaling, likely reflecting the known trafficking properties of these proteins. The crude reintroduction of RAMP1 changed the profile to that of the CGRP receptor, although levels of RAMP expression were likely very high.

Many of the observations in this paper were confirmatory of what was known from cleaner engineered cell systems, but it was reassuring to see this demonstration. The well defined expression profiles and inclusion of appropriate controls made the observations interpretable. The authors do make the case that the current classification of these receptors based exclusively on cAMP responsiveness is likely inadequate to understand the physiology of these systems. It is also reemphasized that the pleiotropic nature of the signaling is ligand-dependent even in this one cellular environment. It may be quite different in other cells expressing this receptor. Of note, this brings new attention to the possible physiological and pathophysiological roles of AM2. The authors also bring attention to other possible influences for these observations, including the composition of the lipid bilayer and other cell specific factors.

The authors provide broad concentration-response curves for the signaling events reported. It would be interesting to also examine different points in time. The temporal contribution to signaling bias is potentially very important and is not highlighted or examined in this report. Some events are quantified at different points in time, but there is no evaluation of whether those time points were the same for each ligand tested. This is potentially quite important. Would the web of bias plots change if

different time points were incorporated?

This provides a clear example of the principles of ligand recognition and signaling to the CGRP receptor family, particularly in endothelial cells. The presentation is clear and fair to point out that this represents only a single example and that many other factors might contribute to in vivo effects of this receptor, and the reader can easily recognize how complex this group of receptors. Accessory proteins, natural ligands might be at all the places these are expressed in vivo. This does make an important contribution to our understanding.

Re: Manuscript COMMSBIO-21-0646-T

30th April 2021

Here in we provide a detailed, point-by-point description in response to the reviewer's comments.

Response to reviewer 1.

This manuscript investigates receptor signalling for the calcitonin gene-related peptide (CGRP) signalling pathway involving the CGRP receptor and those related to it. The study involving primary human cardiovascular cells indicates the ability to influence function and further signalling via biased agonist activity. Should the title also include AM, as this is an important component of the manuscript?

We thank the reviewer for the clear description of our manuscript. We agree with their assessment to change the title to include AM. We therefore propose the new title of the manuscript as:

CGRP, adrenomedullin and adrenomedullin 2 display endogenous GPCR agonist bias in primary human cardiovascular cells

Generally I found navigating around this manuscript, which contains highly relevant and novel data challenging. This is a problem throughout the manuscript.

We apologise for the issues the reviewer has highlighted in navigating around the manuscript. We have endeavoured to improve this within the revised version. To accomplish this, we have rewritten parts of the manuscript to try and highlight the novel results. We have re-ordered some material and moved the description of the generation of CRISPR engineered HUVEC cells to the methods. We hope the changes make the manuscript easier to follow.

The abstract can be improved upon, in that the novelty of the work is not coming through.

We agree with the reviewers' comments regarding the abstract and have rewritten it to try to address the novelty of the work. It should be noted, this is a challenge with the 150 word limit.

Could the authors add a schematic to enable the potential signalling pathways to be more easily understood at the introduction level?

We thank the reviewer for the suggestion of a schematic to enable the signalling pathways to be more easily understood. While we can see the merit of this suggestion, we have included a

schematic in Figure 1 and three further schematics in Figure 5. We would be reluctant to add a further schematic since we feel it might become confusing for the readers.

Results.

1. Can the authors justify the use of human umbilical cells. This is not a typical vascular cell for a number of reasons; although I know later they support the data with cardiomyocytes. Is the result shown in Fig 1/S1 typical for microvascular/aortic sources of human cells from more typical sources too? Note there is a typo in Fig s1 legend line 2 which confuses. Note that not all figures are referred to in Fig 1, before jumping to Fig 2, which makes for a challenge to the reader.

We thank the reviewer for these comments. HUVECs are an established cell line for studying vascular cell processes. Many groups use these cells. We would respectfully highlight that our study also included human aortic endothelial cells (HUAECs) where we observed excellent agreement in signalling profiles with the HUVECs. Furthermore, to enable the CRISPR engineering to be performed, we required a robust cell line that retained its phenotype over multiple passages. HUVECs were ideal for this purpose. Subsequently to submitting the manuscript, and beyond the scope of this present study, we have profiled an additional 10 endothelial cell lines. All show expression of the AM1 receptor alone and share the same pharmacology as the HUVECs and HUAECs. We are confident in our choice of HUVECs and HUAECs for this study.

We have corrected the typo in Figure s1 legend line 2 (now Figure S2). We have also corrected the order of figures as suggested by the reviewer.

2. If Fig S2D is important, why isn't it in the main manuscript?

Here we would respectfully disagree with the reviewer that the figure should be added to the main text. Original Figure S2D showed that intracellular Ca^{2+} mobilisation is mediated in a Gq/11-dependent manner. We do not feel this merits inclusion in the main text. As the reviewer notes, the paper already has a lot of data and we feel adding further to the figures in the main manuscript which are not absolutely necessary will only make it harder to follow the important arguments.

In Fig S2 many of the graphs do not have a letter code. Also is S2E not important for sense of manuscript too. Why is it not in main manuscript?

We apologise for the confusion. We have added additional letters for the code to aid understanding. Again, we respectfully disagree with the reviewer that original figure S2E should be in the main text. It re-enforces the notion of Ca^{2+} -dependent NO release which is well established in the literature.

Can the section on 'system bias' be more fully explained?

We are sorry the reviewer found our description of 'system bias' difficult to understand. We have rewritten the section as:

“Whilst the changes in potency orders seen with cAMP, Ca²⁺, and ERK_{1/2} provide strong evidence for bias, to formally confirm this and to remove potential confounding issue of system bias (which may arise due to the differential expression of signalling components or cofactors in the cellular background of choice) we fitted our HUVEC data with operational model of receptor agonism²⁷ (Figure 1G,H and Table S1). This gives the transducer coefficient $\text{Log}(\tau/K_A)$; effectively the efficacy of an agonist to produce a given response normalised to its functional affinity. To calculate an agonists bias, firstly $\Delta\text{Log}(\tau/K_A)$ is computed; the difference in transducer coefficient for each response compared to AM (Figure 1G) and then the bias factor, $\Delta\Delta\text{Log}(\tau/K_A)$, where there is a second round of normalisation with respect to the cAMP pathway as well as agonist (Figure 1H). This analysis reinforced the notion that AM2 is biased towards Ca²⁺_i mobilisation and NO production, while CGRP favours pERK_{1/2} activation and cell proliferation.”

We hope this clarifies the concept and the calculations.

The exchange of RAMP2 for RAMP1 is clearly documents and nice work. I am slightly confused by the high level of RAMP1. Was it not possible to titrate the transfection level?

We thank the reviewer for the kind comments regarding our studies to exchange the RAMP2 for RAMP1. We agree that the levels of RAMP1 expression are high (which we discuss in the manuscript). We did attempt to titrate the RAMP2 expression, but this unfortunately was not successful. Performing these types of experiments is challenging using primary cell lines. We would also like to highlight that whatever the expression level of RAMP, the surface expression and signalling will be determined by the expression level of CLR, as of course the AM1R is a heterodimer of CLR and RAMP1

I note that similar concentrations of the CGRP antagonist and the AM antagonist were used, but are they equally effective as antagonists?

To clarify regarding the antagonist, olcegepant was used in excess; however, at concentrations of up to 10 micromolar it has no actions on AM receptors (Doods et al., Br. J. Pharmacol., 129 (2000), pp. 420-423). The concentration of AM₂₂₋₅₂ is selective for AM1 receptors (Hay et al., Br J Pharmacol., 2003 Oct;140(3):477-86).

Cardiomyocytes- This data is clear, but the results section ends abruptly, a further line of explanation would be useful.

We thank the reviewer for these comments regarding the cardiomyocyte data. We have added a sentence of explanation to the revised text which reads:

“Overall, our results show that the pattern of bias seen with the CGRP receptor is robust and is transferred between the HUVEC and HCM cell line backgrounds.”

By the end of the manuscript I am impressed by the high quality of the research and potential impact. However I believe the authors should attempt to explain it more easily and in a more

straight forward manner. I think if anything the manuscript is trying to say too much, not helped by the complex nature of this receptor family. Could Fig 5I-L be utilised as a separate Fig for explanation in the discussion.

We thank the reviewer for their kind comments related to the manuscript, their praise for the high-quality research we have performed, and their highlighting of the potential impact of the studies. As noted above, we have moved material to try to make the manuscript easier to read. We have taken onboard the reviewers' comments and reduced Figure 5 so it does not contain the correlation plots (moved to Supplementary Figures). We are now left with Figure 5 containing two bias plots, one plot highlighting CGRP, AM and AM2 potencies and the schematics (originally Figure 5I-L). We hope this better clarifies the data and the message.

I am left wondering about the historical aspects to this study and how this bears on bias agonism. For example, there are a range of studies where various RAMPs have been shown to be raised in certain conditions, for example RAMP1 leading to a supposed upregulation of the CGRP receptor. Additionally, there are historical suggestions that the nature of the receptors may change and additionally that CGRP may be an NO-dependent vasodilator in some vascular tissues, but not others. Could the findings be put into better context with the historical findings?

We thank the reviewer for their suggestion, although we are also mindful of the observation “the manuscript is trying to say too much”. We have included some discussion of previous literature in our consideration of cardiomyocytes. We have added an example into the discussion which now reads as:

“This pleiotropy in signalling pathways is also present in vascular cells, which are themselves far from homogeneous. This may be relevant to the heterogeneity observed in the actions of CGRP and for example, the role of NO or other signalling pathways in vasodilation. Indeed, CGRP has been suggested, in earlier studies, to mediate endothelium-dependent relaxation involving NO via an unknown endothelial receptor⁵⁵. We suggest the AM1 receptor would be a viable candidate for this ‘unknown’ receptor. It is also worth noting that differential changes in RAMP expression have also been observed as a consequence both of pathological changes such as blood pressure and in response to drug treatment; these will have complex effects as a result of bias, also depending on the expression of CGRP, AM and AM2⁵⁶.”

We have added two new references:

55. Gray, D.W, & Marshall, I. Human alpha-calcitonin gene-related peptide stimulates adenylate cyclase and guanylate cyclase and relaxes rat thoracic aorta by releasing nitric oxide. *Br J Pharmacol* **107**, 691–696 (1992)

56. Zhao, Y. et al. Differential expression of components of the cardiomyocyte adrenomedullin/intermedin receptor system following blood pressure reduction in nitric oxide-deficient hypertension. *J Pharmacol Exp Ther.* **316**, 1269-81. (2005).

Line 17 page 3 typo these/there”

This typo has been corrected.

Response to reviewer 2

In this work, the authors explore the ability of ligands to activate the CLR receptor in primary endothelial cells, with detailed insights into signaling pathways and the influence of co-expressed RAMPs. This represents an extension of insights gleaned from cells engineered to express recombinant receptor and RAMPs by focusing on natural cells and on naturally-occurring agonist ligands. This group of receptors and its ligands has already been demonstrated to be significant for the treatment of migraine and has potential importance for cardiovascular disease. Traditionally, the complexity of such a system with multiple receptors, RAMPs, and ligands can be most clearly elucidated taking a reductionist approach in such engineered cell systems, where each variable can be systematically adjusted and the impact determined. By working with primary cells *ex vivo*, the concepts developed previously can be tested in isolated cells. It would be ideal to extend this further to natural tissues and to *in vivo* observations.

Included in this report are two populations of endothelial cells (HUVECs and HUAECs), both of which are characterized to express only CLR and not CTR, and RAMP2 and not other RAMPs (AM1 receptor). Also included for limited observations are primary cardiac myocytes that express only CLR and RAMP1 (CGRP receptor). Each of the cells was carefully evaluated for expression of the relevant genes and for relevant signaling pathways with appropriate positive controls included.

Rank order potency of the natural agonists of the CLR-RAMP2 receptor (AM1 receptor) was as expected, when quantifying cAMP, as has been the tradition. Also, as expected, the signaling responses to these ligands were pleiotropic. The different patterns of signaling events for different ligands, viewed as “bias”, was also clearly demonstrated. In this more natural setting than the engineered cell lines, the varied signaling events for different ligands was nicely demonstrated. It was particularly notable that AM2 was actually more potent than AM at this receptor, but this was dependent on calcium signaling after coupling with Gq/11.

Another interesting line of study was exploration further distally in signaling pathways, including NO and ERK pathways. The evaluation of different agonist effects in these cells provided new insights into what might be possible in mediating cell growth. The case was made that this is likely distinct from the classical focus of this receptor on AM and cAMP.

The authors went further than to only describe these events in the primary endothelial cells, by engineering a change in the RAMP expressed in those cells using gene editing. They showed that the intermediate product lacking RAMP2 exhibited no signaling, likely reflecting the known trafficking properties of these proteins. The crude reintroduction of RAMP1 changed the profile to that of the CGRP receptor, although levels of RAMP expression were likely very high.

Many of the observations in this paper were confirmatory of what was known from cleaner engineered cell systems, but it was reassuring to see this demonstration. The well defined expression profiles and inclusion of appropriate controls made the observations interpretable. The authors do make the case that the current classification of these receptors based exclusively on cAMP responsiveness is likely inadequate to understand the physiology of these

systems. It is also reemphasized that the pleiotropic nature of the signaling is ligand-dependent even in this one cellular environment. It may be quite different in other cells expressing this receptor. Of note, this brings new attention to the possible physiological and pathophysiological roles of AM2. The authors also bring attention to other possible influences for these observations, including the composition of the lipid bilayer and other cell specific factors.

The authors provide broad concentration-response curves for the signaling events reported. It would be interesting to also examine different points in time. The temporal contribution to signaling bias is potentially very important and is not highlighted or examined in this report. Some events are quantified at different points in time, but there is no evaluation of whether those time points were the same for each ligand tested. This is potentially quite important. Would the web of bias plots change if different time points were incorporated?

We thank the reviewer for the encouraging detailed comments regarding the manuscript. We agree with the reviewer about the potential impact of the temporal kinetics when considering agonist bias. As such we have added a note relate to this to the discussion. This reads as:

“It is also important to consider the temporal aspects of signal bias; in this study we have generally measured responses at what seem to be their peak during up to 30-minute stimulations. However, different patterns of bias may be observed if other time points are selected, particularly where signalling from internalised receptors is significant.”

We have compared responses at what we think are likely to be at peak responses (or along the plateau if the response reaches a sustained level, as with cAMP and pERK1/2 – data now added to Supplementary Figure 1). Inevitably we cannot compare identical time points; Ca²⁺ responses are over in 1-2 minutes, proliferation takes several days to occur (see growth rates of cells in Supplementary Figure S4E) and so we note that bias may change with time. However, to fully explore this is beyond our current resources.

This provides a clear example of the principles of ligand recognition and signaling to the CGRP receptor family, particularly in endothelial cells. The presentation is clear and fair to point out that this represents only a single example and that many other factors might contribute to in vivo effects of this receptor, and the reader can easily recognize how complex this group of receptors. Accessory proteins, natural ligands might be at all the places these are expressed in vivo. This does make an important contribution to our understanding.

We thank the reviewer for their support of our work.

Yours sincerely

Dr Graham Ladds FBPhS on behalf of the authors

Reader in Receptor Pharmacology

Tennis Court Road
CB2 1PD

Tel: +44 (0) 1223 333964
Email: gr130@cam.ac.uk

REVIEWERS' COMMENTS:

Reviewer #1 (Remarks to the Author):

Thank you for the revision of this manuscript. I think that it is successful in enabling the reader to be engaged more readily and to understand.

I am happy with the revisions.

Reviewer #2 (Remarks to the Author):

The authors have addressed the issue of temporal changes in various signaling events satisfactorily. The revisions have improved the manuscript. I would have liked to see this extended to natural tissues and to in vivo observations, as requested, but expect this is beyond the scope of the current work.

Re: Manuscript COMMSBIO-21-0646-A

17th May 2021

Here in we provide a detailed, point-by-point description in response to the reviewer's comments.

Response to reviewer 1.

Thank you for the revision of this manuscript. I think that it is successful in enabling the reader to be engaged more readily and to understand.

I am happy with the revisions.

We thank the reviewer for noting the efforts we have made to improve the understanding of the manuscript. We have delighted they are happy with our revisions.

Response to reviewer 2.

The authors have addressed the issue of temporal changes in various signaling events satisfactorily. The revisions have improved the manuscript. I would have liked to see this extended to natural tissues and to in vivo observations, as requested, but expect this is beyond the scope of the current work.

We thank the reviewer for noting our addition of information related to signaling events. We are pleased they thought the manuscript had been improved. We understand their comments related to natural tissues and we hope, in the fullness of time, to archive the aim of validating CLR-based agonist bias in human tissue.

Yours sincerely

Dr Graham Ladds FBPhS on behalf of the authors
Reader in Receptor Pharmacology

Tennis Court Road
CB2 1PD

Tel: +44 (0) 1223 333964
Email: grl30@cam.ac.uk